# Emergence and Transfer of Plasmid-Harbored *rmtB* in a Clinical Multidrug-Resistant *Pseudomonas aeruginosa* Strain

**DOI:** 10.3390/microorganisms10091818

**Published:** 2022-09-11

**Authors:** Jiacong Gao, Xiaoya Wei, Liwen Yin, Yongxin Jin, Fang Bai, Zhihui Cheng, Weihui Wu

**Affiliations:** State Key Laboratory of Medicinal Chemical Biology, Key Laboratory of Molecular Microbiology and Technology of the Ministry of Education, Department of Microbiology, College of Life Sciences, Nankai University, Tianjin 300071, China

**Keywords:** *Pseudomonas aeruginosa*, *rmtB*, multidrug-resistance, conjugative plasmid, mobile genetic element

## Abstract

Multidrug-resistant (MDR) *Pseudomonas aeruginosa* poses a great challenge to clinical treatment. In this study, we characterized a ST768 MDR *P. aeruginosa* strain, Pa150, that was isolated from a diabetic foot patient. The minimum inhibitory concentration (MIC) assay showed that Pa150 was resistant to almost all kinds of antibiotics, especially aminoglycosides. Whole genome sequencing revealed multiple antibiotic resistant genes on the chromosome and a 437-Kb plasmid (named pTJPa150) that harbors conjugation-related genes. A conjugation assay verified its self-transmissibility. On the pTJPa150 plasmid, we identified a 16S rRNA methylase gene, *rmtB*, that is flanked by mobile genetic elements (MGEs). The transfer of the pTJPa150 plasmid or the cloning of the *rmtB* gene into the reference strain, PAO1, significantly increased the bacterial resistance to aminoglycoside antibiotics. To the best of our knowledge, this is the first report of an *rmtB*-carrying conjugative plasmid isolated from *P. aeruginosa*, revealing a novel possible transmission mechanism of the *rmtB* gene.

## 1. Introduction

*Pseudomonas aeruginosa* is a common opportunistic pathogen that widely exists in water, air, human and animal skin, respiratory tracts, and intestinal environments [1,2]. Studies have shown that *P. aeruginosa* usually carries multiple antibiotic resistance genes (ARGs), increasing the difficulties in clinical treatment [3].

Aminoglycoside antibiotics were one of the earliest antibiotics discovered and used clinically. These antibiotics interfere with protein synthesis and ultimately cause bacterial death by binding to the highly conserved A-site of the 16S rRNA of the bacterial 30S ribosomal subunit [4,5]. They have been widely used in the treatment of lung infection caused by *P. aeruginosa* [6,7]. However, resistance to aminoglycosides, including gentamicin, tobramycin, and amikacin, has been increasingly reported worldwide in *P. aeruginosa* [8]. The resistance typically results from low membrane permeability, multidrug efflux systems, mutation of the target gene, and/or chromosome- or plasmid-encoded aminoglycoside modification enzymes (AME) and 16S rRNA methylases (16S-RMTase) [9,10,11].

AMEs inactivate aminoglycosides by modifying their amino or hydroxyl groups, including acetylation (aminoglycoside acetyltransferase [AAC]), phosphorylation (aminoglycoside phosphoryltransferase [APH]), and adenylation (aminoglycoside nucleotidyltransferase [ANT]) [12,13]. 16S-RMTases add a methyl (CH3) group to specific residues of the 16S rRNA, which significantly reduces its binding affinity to aminoglycosides, leading to high-level and extensive aminoglycoside resistance [14]. In recent years, 16S-RMTase has become more and more prevalent in clinical strains [14]. In 2003, a plasmid-mediated 16S rRNA methylase (*rmtA*) was first reported in Japan in a *P. aeruginosa* clinical strain [15], which led to high level resistance to aminoglycoside antibiotics. In recent years, aminoglycoside resistance mediated by an *rmtB* gene carried by plasmids has become a serious threat to almost all aminoglycoside antibiotics [14,16].

Large plasmids in bacteria are mostly conjugative [17,18], and usually carry multiple ARGs that are surrounded by mobile genetic elements (MGEs). Conjugative plasmids play important roles in facilitating the horizontal transfer of ARGs via conjugation [19,20,21,22,23,24]. Many studies have shown that large plasmids carried by *P. aeruginosa* contribute to the accumulation and transmission of ARGs, leading to multidrug-resistance [25,26].

In this study, we characterized an ST768 MDR *P. aeruginosa* strain, Pa150, and identified a 437-Kb plasmid (pTJPa150) containing a 16S rRNA methylase *rmtB*. We identified MGEs flanking *rmtB* and demonstrated the transmission of *rmtB* by the pTJPa150 plasmid through conjugation.

## 2. Materials and Methods

### 2.1. Strains, Plasmids, Primers

The bacterial strains, plasmids, and primers used in this study are listed in Appendix A, respectively.

### 2.2. Antibiotic Susceptibility Test

Overnight bacterial cultures were diluted 1:100 into fresh LB broth to an OD600 of 1. Then, the antibiotics were serially diluted in MHB broth (100 μL) in a 96-well plate. One hundred μL MHB broth containing 10^6^ CFU/mL cells were added to each well with the diluted antibiotics. The plates were incubated at 37 °C for 24 h. The MICs were determined by the lowest antibiotic concentration with no visible bacterial growth.

### 2.3. Whole Genome Sequencing (WGS) and Bioinformatics Analysis

Total DNA from Pa150 was extracted using the TIANamp Bacteria DNA Kit (TIANGEN, Beijing, China) according to the manufacturer’s instructions. DNA library construction and sequencing were performed by Grandomics Biosciences Co., Ltd., Beijing, China. Single-molecule sequencing was performed with a PromethION sequencer (Oxford Nanopore Technology, Bejing, China). Genome assembly, correction, and optimization were performed on the obtained data to obtain the final genome. CGView server software was used for plasmid circle mapping and MEGA software (version 7.0, Auckland, New Zealand) was used for evolutionary analysis.

### 2.4. Gene Cloning and Conjugation Assays

A DNA fragment containing the *rmtB* gene with its native transcription terminator region was amplified by PCR. The PCR product was digested with *Bam*HI and *Eco*RI, and then cloned into the pUCP20 vector to yield the recombination plasmid pUCP20-*rmtB*. The resultant plasmid was transferred into *P. aeruginosa* reference strain PAO1 via electroporation.

The conjugation assay was performed by using a tetracycline-resistant mutant of PAO1 carrying a *lacZ* gene (PAO1-*lacZ*) as the recipient strain [27,28]. The donor strain Pa150 and the recipient strain PAO1-lacZ were grown in LB broth to an OD600 of 0.8. The donor bacteria and the recipient bacteria were mixed at a ratio of 7:1. The bacteria were centrifuged at 6000 rpm for 5 min and then washed three times with 1 mL fresh LB broth. Then, 100 μL of the bacterial suspension were added onto a filter membrane, which was placed on a Nutrient-Agar (BD, Difco) plate at 37 °C for 12 h. The bacteria were washed out from the membrane by LB and plated on LB agar plates containing X-Gal (40 mg/mL), tetracycline (150 μg/mL), chloramphenicol (200 μg/mL), and isopropyl β-D-1-thiogalactopyranoside (IPTG, 20 μg/mL). Blue colonies were further streaked on the selection plates and the transconjugants were verified by PCR with the primes listed in Appendix A.

### 2.5. Statistical Analysis of All Strains Containing the rmtB Gene

A basic local alignment search tool (BALST) was used for searching sequences sharing high similarity (>90% identity) with the *rmtB* gene. Information about all strains (*n* = 620) containing the *rmtB* gene was collected, including species, isolation years, sources, and countries (Appendix A). The analytic results were presented by line, pie, and bar charts using the Prism software (Graphpad).

### 2.6. Data Availability

The genome and plasmid sequences were deposited in the NCBI Genbank database (CP094677, CP094678)

## 3. Results

### 3.1. Characterization of MDR P. aeruginosa Strain Pa150 and Conjugative Plasmid pTJPa150

A *P. aeruginosa* strain (named Pa150) was isolated from a diabetic foot patient in Baoding, Hebei Province, China. This isolate is resistant to almost all classes of antibiotics (Table 1), including meropenem (MEM, MIC = 64 μg/mL), aztreonam avibactam (AZT, MIC = 64 μg/mL), azithromycin (AZM, MIC = 128 μg/mL), ciprofloxacin (CIP, MIC = 32 μg/mL), amikacin (AMK, MIC > 128 μg/mL), and tobramycin (TBR, MIC > 128 μg/mL), but remains susceptible to colistin (COL, MIC = 0.25 μg/mL). Whole genome sequencing (WGS) of Pa150 was performed (Figure 1). The genome of Pa150 was around 6.6 Mb, with GC content of 65.62% (Appendix A), encoding 6497 genes. Multilocus sequence typing (MLST) revealed that Pa150 belongs to ST768. ST175, ST235, ST357, and ST664 are high risk clones of P. aeruginosa, with the highest clinical isolation rates in the world [29,30,31]. However, the MDR ST768 clone is rarely reported.

Notably, Pa150 harbors a large conjugative plasmid, pTJPa150. The pTJPa150 plasmid is 436,716 bp in size with a GC content of 56.85%; it contains 464 predicted open reading frames (ORFs) (Figure 2). The pTJPa150 plasmid has the highest coverage and sequence similarity (93% + 99.14%) with the plasmid pBT2436 of the *P. aeruginosa* strain T2436 (GenBank accession number CP039989), which was isolated from adult male sputum in Thailand in 2013. Further sequencing analysis revealed that the pTJPa150 plasmid contains multiple ARGs, including two β-lactamase genes (*bla_TEM_* and *bla_OXA-10_*), five aminoglycoside resistance genes (*aac(6′)-IIa*, *aac(3′)-IId*, *strA*, *strB*, and *rmtB*), a single-component efflux pump gene *cmlA* and a resistance-nodulation-cell-division (RND) pump gene cluster *tnfxB-tmexCD-toprJ* (Appendix A), indicating a potential role in the multidrug-resistance of Pa150.

### 3.2. Contribution of the Conjugative Plasmid pTJPa150 in Antibiotic Resistance

The conjugative transfer of bacterial plasmids is the most effective mode of horizontal gene transmission and is, therefore, considered to be one of the main causes for the increase in multidrug-resistant bacteria [18,32]. In Gram-negative bacteria, conjugation is usually mediated by the conjugative type IV secretion system (T4SS), a large transport apparatus produced by donor cells [33,34]. The pTJPa150 plasmid contains genes involved in conjugation (Figure 2). To verify the self-transmissibility of the pTJPa150 plasmid and its role in the MDR of Pa150, we performed a conjugation assay by using Pa150 and a *lacZ* expressing *P. aeruginosa* reference stain PAO1 (PAO1-*lacZ*) as the donor and recipient strains, respectively. The presence of the pTJPa150 plasmid in the transconjugants was verified by PCR amplification of the *bla_TEM_* and *parB* genes. Compared with the recipient strain, the MICs of β-lactam, macrolide, and quinolone antibiotics for the transconjugant were two- to eight-fold higher (Table 1). Particularly, the MICs of aminoglycoside antibiotics were increased by more than 128-fold, reaching the similar levels of Pa150 (Table 1), indicating a significant role of the pTJPa150 plasmid in aminoglycosides resistance. The treatment of Pa150 and the transconjugant (PAO1-*lacZ*-pTJPa150) with the RND pump inhibitor PAβN reduced the MICs of the β-lactam, macrolide, and quinolone antibiotics, but had no effect on the MICs of aminoglycosides (Table 1). These results demonstrate a role of TMexCD-TOprJ in the resistance to β-lactam, macrolide, and quinolone antibiotics, and additional mechanisms in the resistance to aminoglycosides.

### 3.3. Characterization of the Role of RmtB in Aminoglycoside Resistance

As shown in Figure 2 and Appendix A, the pTJPa150 plasmid contains four aminoglycoside modification enzyme genes (*aac(6′)-IIa*, *strA*, *strB*, *aac(3′)-IId*) and a 16S rRNA methylase gene *rmtB*. Previous reports have shown that AAC(6′) enzymes are the most common AMEs that are widely distributed in Gram-negative and Gram-positive bacteria. However, AAC(6′)-II enzymes cannot modify amikacin. AAC(3)-II enzymes are almost exclusively found in Gram-negative bacteria and confer resistance to many antibiotics, including gentamicin, neomycin, tobramycin, and sisomicin [13]. *strA* and *strB* are currently the most widely distributed streptomycin resistance determinants. However, these four AMEs are rarely reported to contribute to high amikacin resistance [13,35,36]. RmtB was first identified in *Serratia marcescens*. It shares an 82% amino acid sequence identity with RmtA and has been shown to lead to high-level resistance to various aminoglycosides [37]. To verify the role of *rmtB* in the aminoglycoside resistance, we cloned the gene into *E. coli* DH5α and PAO1. The expression of *rmtB* significantly increased the MICs of the aminoglycosides in both strains (Table 1), indicating an important role of *rmtB* in the aminoglycoside resistance.

### 3.4. Possible Origin and Mobilization of rmtB

On the pTJPa150 plasmid, *rmtB* is in close proximity to *bla_TEM_*, flanked by multiple MGEs (Figure 2). Phylogenetic analysis revealed that RmtB has high amino acid homology in 53 phylogenetic groups (90%) (Figure 3A), mainly including five *P. aeruginosa* isolates (green), 15 *E. coli* isolates (blue), and 28 *Klebsiella pneumonia* isolates (orange). These *rmtB*-carrying strains were isolated from various countries since 2005, including the USA, China, Thailand, Vietnam, and Canada. In addition, the amino acid sequence of RmtB in the pTJPa150 plasmid is more closely related to that of the *K. pneumoniae* HS11286 plasmid pKPHS3.

Like the reported *rmtB* gene in *S. Marcescens*-95, the *rmtB* gene in the pTJPa150 plasmid is flanked by *bla_TEM_* and Tn*3*. Additionally, IS*91*, IS*30*, and IS*26* transposons and the *rmtB*-*bla_TEM_*-Tn*3* cluster form an IS*91*-*groEL*-*nahP*-*rmtB*-*bla_TEM_*-Tn*3*-IS*30*-IS*26* module (Figure 3B). Sequence alignment using the *rmtB-bla_TEM_-*Tn*3* cluster sequence revealed a total of 520 sequences with homology greater than 99% and query cover greater than 90% in the NCBI database. We then selected the top 11 sequences with the highest homology to the IS*91*-*groEL*-*nahP*-*rmtB*-*bla_TEM_*-Tn*3*-IS*30*-IS*26* module in Pa150 as the representatives to show the alignments (Figure 3B). *K. pneumoniae* HS11286, the closest strain to Pa150 in the phylogenetic tree, was included in the 11 strains. Through linear sequence alignment, we found that there were insertion sequence (IS) elements on both sides of the *rmtB*-*bla_TEM_*-Tn*3* cluster in all of the 11 strains. Of the 11 strains, the MGEs on both sides of the *rmtB*-*bla_TEM_*-Tn*3* cluster were IS*26* elements (Figure 3B), except for the plasmid pKPHS3 in *K. pneumoniae* strain HS11286 and the plasmid pHN21SC92-1 in *Citrobater Portucalensis* strain GD21SC92T (Figure 3B). Wang et al. found that IS*26* elements play a vital role in the dissemination of the resistance genes (16). However, in the pTJPa150 plasmid, the *rmtB*-*bla_TEM_*-Tn*3* cluster is flanked by IS*26* and IS*91*, which is the same in the *C. portucalensis* plasmid pHN21SC92-1. Further analysis revealed that part of the IS*91* sequence showed 100% identity with the IS*CR15b* in the *K. pneumoniae* plasmid pKPHS3 (Figure 3B), suggesting the transfer of the gene cluster in this plasmid may also be mediated by IS*26* and IS*CR15b*.

### 3.5. Epidemic Analysis of rmtB-Carrying Strains

To date, 620 *rmtB*-carrying strains have been reported worldwide. Among all of the strains, only 1.77% (*n* = 11) are *P. aeruginosa*; the majority are *K. pneumonia* (54.03%) and *E. coli* (34.19%). In 90.64% of these strains, *rmtB* is present on plasmids (Figure 4A, Appendix A), 90% of which are larger than 10 Kb. Given the important roles of plasmids in the transmission of resistance genes [38], the presence of *rmtB* on plasmids may account for its worldwide pread. In all of the analyzed samples, the most frequent isolation source was the clinical environment (*n* = 355), followed by animal sources (*n* = 60) and natural environments (*n* = 25) (Figure 4B, Appendix A). It is worth mentioning that these *rmtB*-carrying strains are almost all pathogenic bacteria.

From another perspective, these *rmtB*-carrying strains were isolated from 32 different countries, with the largest total number of strains isolated from China (Figure 4D, Appendix A). From 2003 to 2022, *rmtB*-carrying strains were isolated from 24 provinces or regions in China; they were highly prevalent in Zhejiang (17.9%), Guangdong (15.2%), Sichuan (10.22%), and Henan (11.97%) provinces (Figure 4C, Appendix A). According to the dataset, *rmtB* was first reported in 2003, and the reported number of *rmtB*-carrying strains kept increasing until 2017, then decreased to 47 in 2021 (Figure 4E, Appendix A).

## 4. Discussion

*P. aeruginosa* is a common clinical opportunistic pathogen and one of the main sources of nosocomial infection [39]. With the increase in clinical MDR strains, the World Health Organization (WHO) has listed MDR *P. aeruginosa* as one of the most threatening human pathogens [40]. In this study, we identified an MDR *P. aeruginosa* strain, Pa150, that contain a conjugative plasmid, pTJPa150. The pTJPa150 plasmid harbors an *rmtB* gene, which contributes significantly to aminoglycoside antibiotic resistance (Table 1). As far as we know, this is the first report of an *rmtB*-carrying plasmid in *P. aeruginosa*.

Plasmid conjugation is a major route in the spread of ARGs [41]. In this study, we found that the transconjugant carrying the pTJPa150 plasmid displayed elevated MICs of β-lactam, quinolone, and aminoglycoside antibiotics, compared with the parental recipient strain (PAO1-*lacZ*) (Table 1). In the pTJPa150 plasmid, a large number of ARGs form gene clusters at specific locations (Figure 2). Site-specific recombinases and transposases that may mediate horizontal transfer also exist in the pTJPa150 plasmid and are located close to the ARGs (Figure 2). The *rmtB* gene was found in the pTJPa150 plasmid containing multiple MGEs, which might promote the spread of *rmtB* in *P. aeruginosa*. In addition, the pTJPa150 plasmid contains an RND efflux pump *tnfxB-tmexCD-toprJ*. The pump has been identified on plasmids in *P. aeruginosa* and *K. pneumonia*, which significantly increased the bacterial resistance to tigecycline [42,43]. Further studies are warranted to understand its role in bacterial resistance to various types of antibiotics.

AMEs and 16S rRNA methylase are the main causes of bacterial resistance to aminoglycoside antibiotics. Gram-negative bacteria containing *rmtB* also carry other enzymes that are responsible for different kinds of antibiotic resistance, such as β-lactamase [44]. In this study, the nucleotide sequences of *rmtB* adjacent to *bla_TEM_* in the pTJPa150 plasmid was 100% identical to that in pS95B2 in *S. marcescens* S-95, which was first identified as the *rmtB*-carrying strain [27]. It has been reported that *rmtB* in *K. pneumoniae* might be transferred through an IS*26*-mediated homologous recombination, although no IS*26* recognition repeat sequence has been found [16]. In the pTJPa150 plasmid, the IS*26* sequence exists only on one side of *rmtB*, while on the other side is IS91, part of which is actually an IS*CR15b* sequence. Wang et al. also showed the diversity of MGEs on both sides of the *tet(X6)* gene [43]. Based on this observation and the finding in this study, we speculated that this diversity might be more conducive to the transfer of resistance genes. Strains containing the *rmtB* gene are widely distributed worldwide and have been isolated in 32 countries. Almost all the known *rmtB*-carrying strains are Enterobacteriaceae strains. Most of the *rmtB* genes exist on plasmids, which may be an important reason for the rapid spread of the *rmtB* gene since its discovery in 2003. In addition to clinical samples, edible animals or animal feeding environments are also the main sources of the *rmtB*-carrying strains of the reported 620 strains, which may promote the transfer of *rmtB* to humans through the food chain.

In summary, we reported for the first time an *rmtB* gene in a plasmid of an MDR *P. aeruginosa* strain. The transfer of the *rmtB* gene might be mediated by IS*26* and IS*CR15b*. In addition, we carried out statistical analysis on all strains containing the *rmtB* gene and found that *rmtB*-carrying strains are widely distributed throughout the world. Most of them exist on the plasmids in the strains.

## Figures and Tables

**Figure 1 microorganisms-10-01818-f001:**
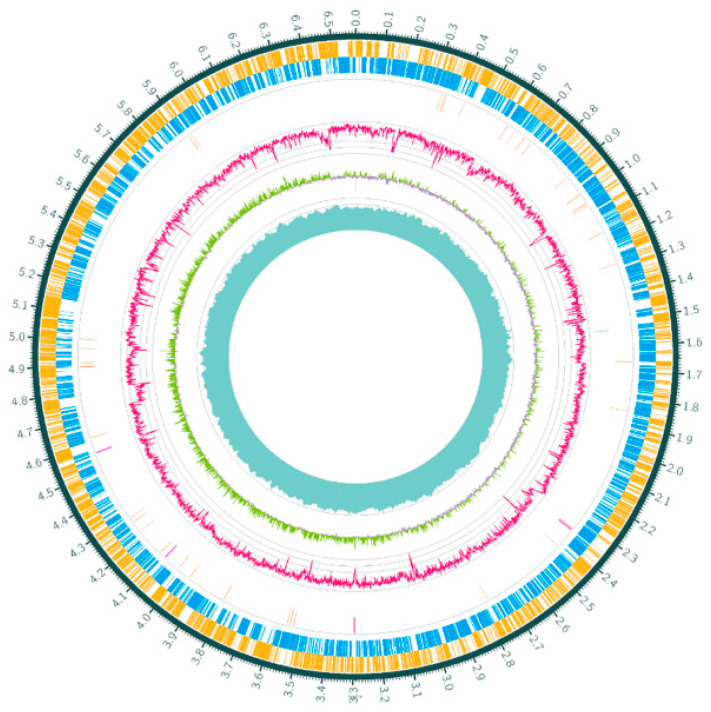
Graphical circular schematic map of the clinical isolate Pa150 from circle 1 (outmost) to circle 7 (innermost). Circles 1 and 2, coding sequences (CDS), forward and reverse frames, respectively; circle 3, tRNAs (orange) and rRNAs (purple); circle 4, CRISPR-related genes (blue); circle 5, GC content; circle 6, GC-Skew; circle 7, coverage.

**Figure 2 microorganisms-10-01818-f002:**
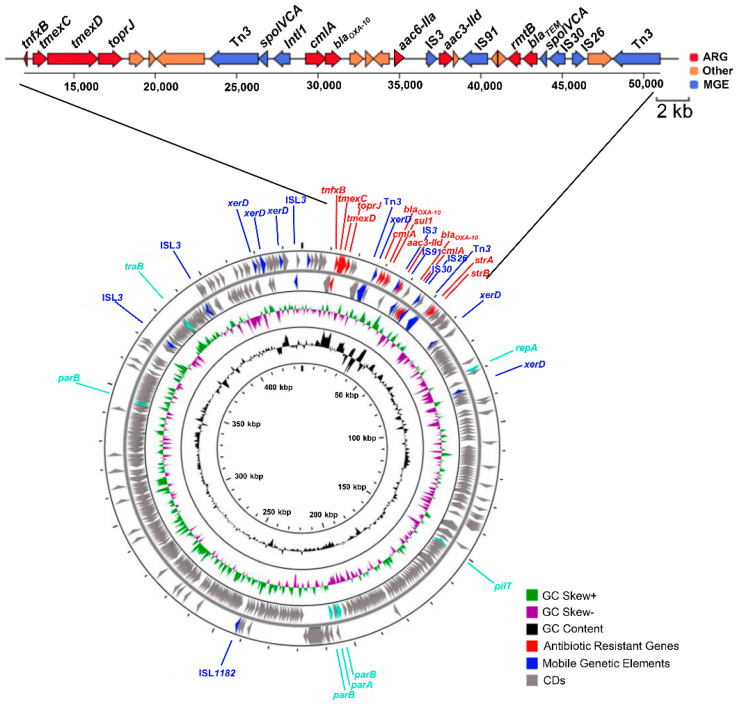
Schematic map of the pTJPa150 plasmid. The innermost circle is the scale. The GC content is illustrated in the second circle. The third circle shows GC skew (+: green, −: purple). The outermost circle indicates the CDs, in which antibiotic resistance genes are highlighted in red and mobile genetic elements in blue.

**Figure 3 microorganisms-10-01818-f003:**
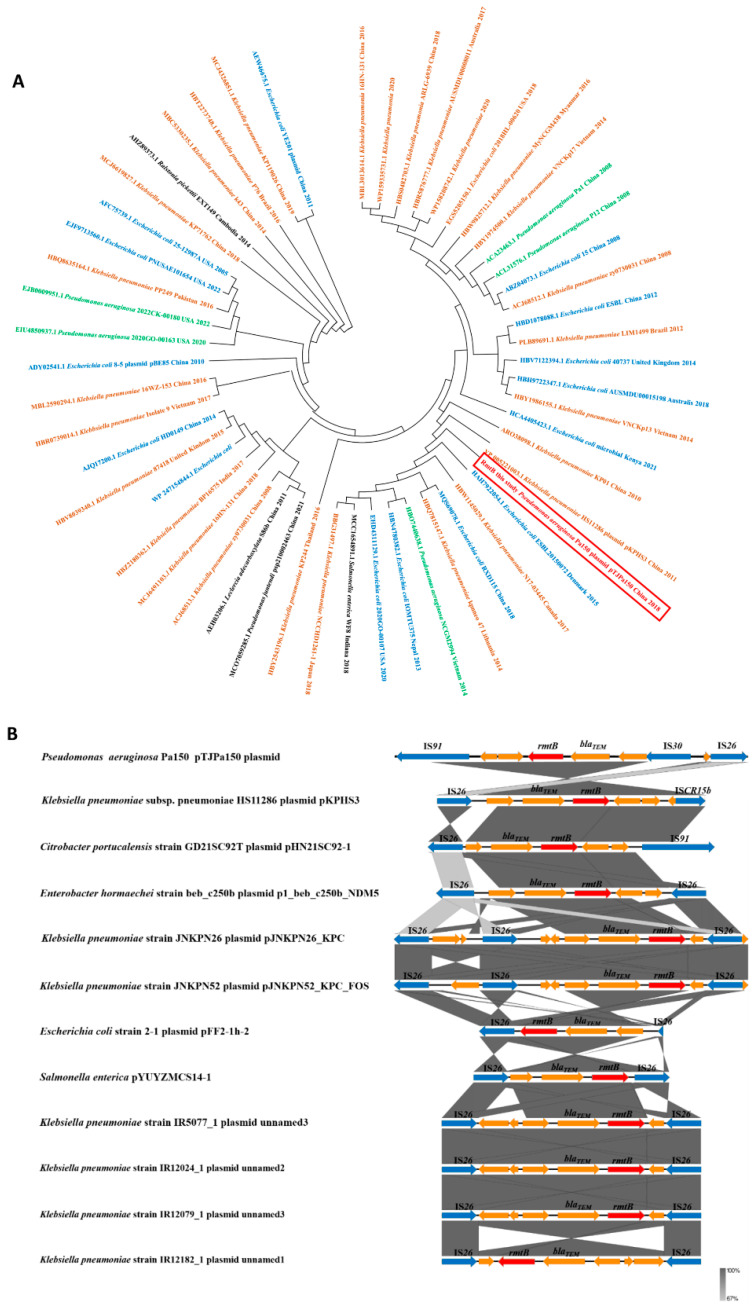
Phylogenetic tree and genetic information of *rmtB*. (**A**) Phylogenetic tree of RmtB and RmtB-like proteins. Neighbor-joining tree based on the amino acid sequences of RmtB and RmtB-like proteins (obtained from the NCBI databases) generated using MEGA7; the bootstrap was 1000. The bacteria species are indicated by the colors as follows: *P. aeruginosa*, green; *K. pneumonia*, orange; *E. coli*, blue; others, black. (**B**) Comparison of the genetic environment of *rmtB* with those of closely related sequences. The mobile genetic elements are shown in blue.

**Figure 4 microorganisms-10-01818-f004:**
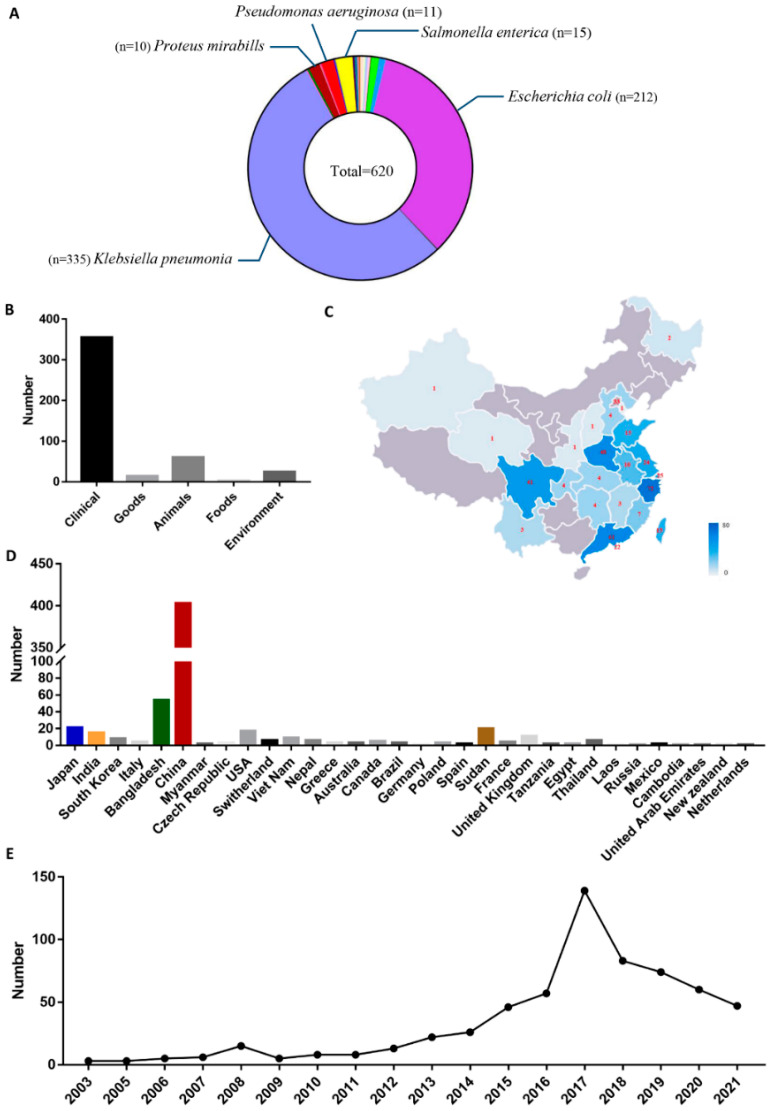
Prevalence and distribution of *rmtB*-positive isolates. (**A**) Species classification of *rmtB* positive isolates. (**B**) Source statistics of *rmtB*-positive isolates (**C**) Geographical distribution of *rmtB*-positive isolates in China. The color depth correlates with the quantity. (**D**) Distribution of *rmtB*-positive isolates worldwide. (**E**) Isolation of *rmtB*-positive isolates from 2003 to 2021.

**Table 1 microorganisms-10-01818-t001:** MICs (μg/mL) for indicated strains.

Strain	Strain Information	MEM	AZT	CTZ	AZT +AVI	CTZ +AVI	AZM	CIP	TBR	GEN	AMK	COL
Pa150	Isolate from Hebei	64	64	64	64	16	128	32	>128	>128	>128	0.25
Pa150+PAβN		8	8	2	16	2	16	4	>128	>128	>128	
PAO1-*lacZ*	PAO1 expressing *lacZ*	0.5	4	2	2	4	128	0.125	0.25	0.5	1	
PAO1-*lacZ*+PAβN		0.5	1	0.5	2	0.5	8	0.125	0.25	0.5	1	
PAO1-*lacZ*-pTJPa150	PAO1 conjugates Pa150 plasmid	1	16	8	16	8	>128	0.25	>128	>128	>128	
PAO1-*lacZ* pTJPa150+PAβN		0.5	0.5	0.5	2	1	16	0.0625	>128	>128	>128	
DH5α-pPUCP20	DH5α transformants with the empty vector								0.5	0.25	0.125	
DH5α-pPUCP20-*rmtB*	DH5α transformants expressing *rmtB*								>128	>128	>128	
PAO1-pPUCP20	PAO1 transformants with the empty vector								1	0.5	0.25	
PAO1-pPUCP20-*rmtB*	PAO1 transformants expressing *rmtB*								>128	>128	>128	

MEM, meropenem; AZT, aztreonam; CTZ, ceftazidime; AVI, avibactam; AZM, azithromycin; CIP, ciprofloxacin; TBR, tobramycin; GEN, gentamicin; AMK, amikacin; COL, colistin.

## Data Availability

Not applicable.

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
