# Peer review of "Emergence and Transfer of Plasmid-Harbored rmtB in a Clinical Multidrug-Resistant Pseudomonas aeruginosa Strain"

_microorganisms, 2022, doi:10.3390/microorganisms10091818_

Round 1

Reviewer 1 Report

I have read the manuscript entitled "Emergence and transfer of plasmid-harbored rmtB in a clinical multidrug-resistant Pseudomonas aeruginosa strain" with great interest and I think it is in principle suited for a publication in the Microorganisms, Special Issue: Plasmids Carrying Antimicrobial Resistance Genes in Gram-Negative Bacteria. The authors characterized a new P. aeruginosa strain (Pa150) and identified a 437 kb plasmid (pTJPa150) containing 16S rRNA methylase gene rmtB. Overall this is a well conducted study and the results were presented clearly. I feel this manuscript has a lot of value to readership and the paper brings some new information to the scientific community. I am happy that references are appropriate, that tables and figures are well presented. Nevertheless, I have some concerns, and several minor changes are required.

Minor comments:

Lines 95-96: “The genome and plasmid sequences were deposited in the NCBI Genbank database  (CP094677, CP094678)”. Please, check the data. I didn't find CP094677 and CP094678 in the database.

Lines 253-254: “In addition, we carried out statistical analysis on all strains containing the rmtB gene…” You need to explain the statistical analysis and add it’s in the section “Materials and Methods”.

Please, improve the quality of the Figure 2.

Table 1. Please, add an explanation for the abbreviations (MEM, AZT, CTZ, …)

Best regards.

Author Response

  1. Lines 95-96: “The genome and plasmid sequences were deposited in the NCBI Genbank database (CP094677, CP094678)”. Please, check the data. I didn't find CP094677 and CP094678 in the database.

Response: We contacted with the NCBI database and the data has been released.

  1. Lines 253-254: “In addition, we carried out statistical analysis on all strains containing the rmtB gene…” You need to explain the statistical analysis and add it’s in the section “Materials and Methods”.

Response: We have added the detailed selection method of the 620 strains containing the rmtB gene in the section 2.5.

  1. Please, improve the quality of the Figure 2.

Response: Thanks for your suggestion. Figure 2 has been changed to a clearer figure now.

  1. Table 1. Please, add an explanation for the abbreviations (MEM, AZT, CTZ, …)

Response: Thank you for the suggestion and we added the related explanation for the abbreviations to the footnote.

Reviewer 2 Report

The authors describe the presence of a plasmid containing rtmB in P. aeruginosa, a feature not described before. The plasmid contains several other antibiotic resistance genes, inlcuding a tripartite MDR efflux pump. These RND efflux pumps are rarely described to be plasmid-located and hence this finding should be discussed in more detail. 

The section 3.5 is poor. The authors would like to compare the genetic structures surrounding rtmB in all cases where sequences are available. This analysis was done in a restricted number of isolates in 3.4. However in an era of WGS, all available sequences should be used.

The authors must include a legend to explain what Figure 1 shows. 

Author Response

  1. The authors describe the presence of a plasmid containing rtmB in P. aeruginosa, a feature not described before. The plasmid contains several other antibiotic resistance genes, inlcuding a tripartite MDR efflux pump. These RND efflux pumps are rarely described to be plasmid-located and hence this finding should be discussed in more detail.

Response: Thanks for the suggestion. We added more detailed information about RND efflux pumps to the Discussion part (line 253-257 in the revised version).

  1. The section 3.5 is poor. The authors would like to compare the genetic structures surrounding rmtB in all cases where sequences are available. This analysis was done in a restricted number of isolates in 3.4. However in an era of WGS, all available sequences should be used.

Response: Thank you for pointing this out. We have further enriched and improved section 3.5. There are indeed a lot of available sequences with a total of 520 sequences with homology greater than 99% and query cover greater than 90% in the NCBI database aligned with the rmtB-blaTEM-Tn3 cluster sequence. We then selected the top 11 sequences with the highest homology to the IS91-groEL-nahP-rmtB-blaTEM-Tn3-IS30-IS26 module in Pa150 as the representatives to show the alignments. The description has been added to the manuscript.

  1. The authors must include a legend to explain what Figure 1 shows.

Response: As requested, we added the description of the genome circles in the figure legend (line 117-120 in the revised manuscript).

Reviewer 3 Report

In this article by Gao et al., the authors have characterized a multidrug resistant clinical isolate of Pseudomonas aeruginosa named Pa150. They used whole genome sequencing to report the presence of multiple antibiotic resistance chromosomal genes and a 16S rRNA methylase gene (rmtB) carrying conjugative plasmid (pTJPa150) in the bacteria. The authors also demonstrated a role of RmtB in the generation of aminoglycoside resistance in Pa150. Finally, they suggest a possible mechanism for the transmission of this highly prevalent MDR gene in P. aeruginosa. The study is well-conducted, and conclusions supported by the data.  The article would appeal to the readers interested in understanding the various mechanisms of drug resistance amongst pathogenic bacteria. A few minor comments are:

1. The labeling in Figure 2 is illegible. Consider magnifying the circular schematic map of the plasmid with larger fonts.

2. Line 130: Multiple drug-resistant bacteria or multidrug resistant bacteria

Author Response

  1. The labeling in Figure 2 is illegible. Consider magnifying the circular schematic map of the plasmid with larger fonts.

Response: Thanks for your suggestion. We have replaced the figure with a larger one with higher resolution.

  1. Line 130: Multiple drug-resistant bacteria or multidrug resistant bacteria

Response: Thank you for pointing this out. We have changed it into multidrug-resistant bacteria.

Round 2

Reviewer 2 Report

The authors have properly addressed my queries